# Competition in *Abies kawakamii* forests at subtropical high mountain in Taiwan

**Wei Wang[1,2☯], Min-Chun Liao [2☯], Hsy-Yu Tzeng[1] ***

**1** Department of Forestry, National Chung Hsing University, Taichung City, Taiwan (R.O.C.), **2** Botanical Garden Division, Taiwan Forestry Research Institute, Taipei City, Taiwan (R.O.C.)

☯ These authors contributed equally to this work.
* erecta@nchu.edu.tw

**Data Availability Statement:** All relevant data are within the manuscript and its Supporting Information files.

**Funding:** This research funding was supported by Shei-Pa National Park in Taiwan. The funders had

## Abstract

The spatial patterns of plant species reflect the competitive pressures on individuals. To generate Competition indices (CI), we measured the diameter at breast height (DBH), crown volumes (CV) and the distances between trees. In this study, *Abies kawakamii* were divided based on the dominant component of the understory (moss or bamboo) to (1) investigate the relationship between the CI and stand structural attributes (SSAs); (2) compare the inter- and intraspecies; CIs as well as living and dead individual CIs; and (3) examine the relationship between the DBH and CI. The current findings indicate that the understory composition affected the CI and SSAs. The interspecies CI was larger than the intraspecies CI when bamboo-dominated the understory. In contrast, the intraspecies CI was larger than the interspecies CI when the understory was dominated by moss. The CI of dead individuals was higher than that of live individuals due to the biological characteristics and regeneration needs of *Abies*. Additionally, sensitivity to the environment and available resources may exert more pressure on young individuals than mature individuals.

## Introduction

Tree distribution and density depend on environmental conditions in the forest [1, 2]. Forest communities are shaped by spatial variation [3, 4], seed spread, interspecies and intraspecies competition, heterogeneity of the environment, and different scales of variability that affect spatial patterns [5]. In the harsh environment of high-altitude forests, species may interact competitively or symbiotically. Therefore, investigating the relationships among adjacent trees of different sizes and species in forests at various spatial scales will allow us to better understand their coexistence mechanism. The competitive relationships between individual trees can be incredibly complex, reflecting the niches of individuals of various species and their resource and space needs [6, 7].

Competition indices (CI) are used by researchers in many fields to quantify various attributes of competition among individuals or groups [8–10]. Forest CI may include SSAs, such as biomass, crown height, crown length, and diameter at breast height (DBH) [10]. Indices allow researchers to condense and organize experimental results, facilitating the interpretation of

no role in study design and analysis, decision to publish, or preparation of the manuscript.

**Competing interests:** The authors have declared that no competing interests exist.

complex datasets as well as enabling comparisons among different studies [11]. Many forest ecology management studies use CI that incorporate coexistence theory [12, 13]. Furthermore, some studies have indicated that if one population's intraspecific competition is more significant than its interspecific competition, the coexistence effect replaces the competition effect [14, 15]. Based on these studies, we aimed to determine the competition effect of a stable conifer population in Taiwan.

*Abies kawakamii* is an endemic conifer species and glacial relict in Taiwan; it is the primary species populating the subalpine forest line. Other Abies species are typically distributed in frigid and temperate zones, but *A. kawakamii* is distributed in subtropical areas; thus, *A. kawakamii* reflects the ecological distribution of this genus at its southern extremes. *A. kawakamii* is continually distributed from north to south at altitudes of 2,400–3,600 m, primarily on Taiping Mountain, Nanhu Mountain, and Peinan Mountain, Taiwan [16]. *A. kawakamii* often comprises single-species stands in subalpine regions, demonstrating a simple and representative stand structure in the alpine ecosystem. The most famous *A. kawakamii* forest site in Taiwan is in the Mt. Xue black forest. Although several studies have examined Taiwan fir forest structures [16–20], the coexistence mechanism between Taiwan fir populations and other species is still unclear.

In this study, we explore the stand structure attributes (SSAs) to discuss the relationship with CI [21]. We set up sampling plots on Mt. Xue to examine the *A. kawakamii* forest structure in this area. Based on the proposed competition index, we measured the DBH of each individual and the distances between individuals. Furthermore, we separated the dataset sources from species and health status to discuss the ecological means [6, 7, 9, 22]. In this study, we evaluated the competitive relationships and structures of *A. kawakamii* on Mt. Xue to elucidate changes in the construction and spatial patterns of the *A. kawakamii* population. We assume five hypotheses that (1) CI based on variable distance would search more competitor trees than CI based on fixed distance; (2) CI would be positive corelated with SSAs; (3) Individuals of Taiwan fir would be sustain more competitive stress from intraspecies than interspecies; (4) Dead individuals would be sustain more competitive stress than living individuals; (5) There were negative relationship with CI and DBH of target trees.

## Materials and methods

### Study sites

The study sites A1-A7 were located at a long-term ecological research site (est. 2009) in Shei-Pa National Park, northern Taiwan (Fig 1) [17]. Shei-Pa National Park permitted us to collect plants. The sites were located along the trail to Mt. Xue (3,886 m), the highest peak (24˚ 23'0.24"N, 121˚13'54.48"E). Climate data have been collected at the Mt. Xue alpine weather station (3,590 m) since 2009. The mean annual temperature is 9.9˚C, and the mean annual rainfall is 2,774.2 mm at the study site (Fig 2). The minimum temperature recorded was −6.1 in January, and the maximum temperature was 21.2˚C in July. In the winter of 2009–2010, 34 snow flurry activities were recorded. The climate in the region is classified as subtropical subalpine humid (Thornthwaite climate type AC'$_1$ra) [23]. The Mt. Xue study area geological region is in the western subregion of the Central Mountains, which features tertiary sub-metamorphic rock composed of dark gray argillite and slate rocks [24]. The soil is very acidic (pH 4.5) and phosphorus-deficient, with a high exchangeable base of aluminum and rock content rate of approximately 10% [25].

Mt. Xue forests are dominated by *A. kawakamii*, *Juniperus squamata* var. *morrisonicola*, *Tsuga chinensis* var. *formosana*, and *Sorbus randaiensis*. Seven plots (40 × 50 m$^2$ [0.2 ha]) were established in 2008 and reassessed in 2018 for this study. Plot locations were accessible and

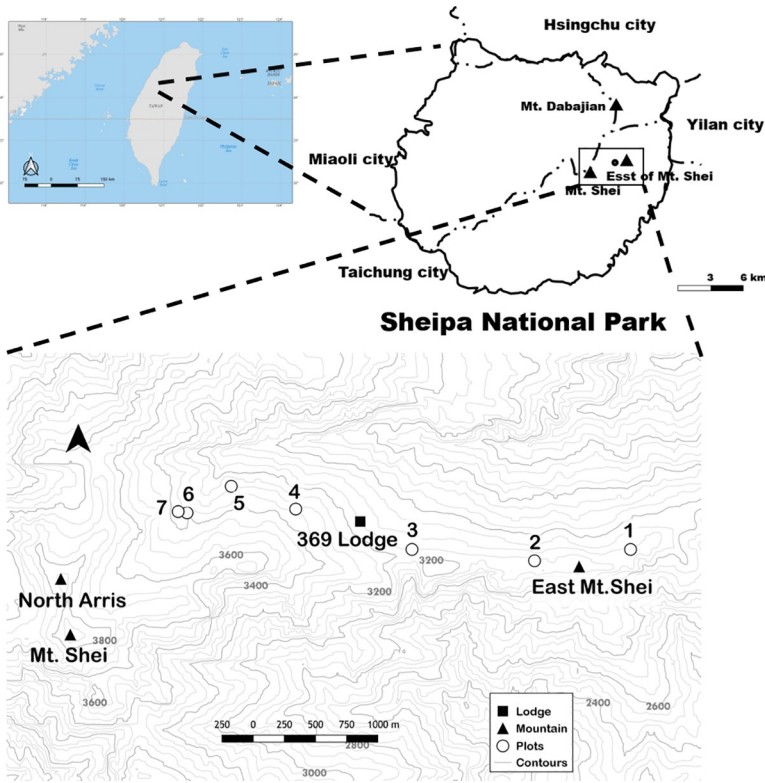

**Fig 1. Seven plots in the study area on Mt. Xue in Taiwan.** Plots were drawn in QGIS 3.4 using data from the Natural Earth Quick Start Kit (https://www.naturalearthdata.com/downloads/) and Government Information Open Platform in TW (The area of Shei-Pa National Park: https://data.gov.tw/dataset/46541; DEM of Taiwan: https://data.gov.tw/dataset/35430). This original content was made in this study under the CC BY 4.0 license.

representative of black forests of uneven ages at 3,100–3,350 m in 2008 [17]. All living trees taller than 1.3 m were mapped, tagged, and measured (DBH and height). The seven plots encompassed various stand characteristics, minimizing the chance of drawing unreliable inferences based on a small plot sample (Table 1) [17]. We categorized the plots based on understory compositions of moss or bamboo (Yushan cane, *Yushania niitakayamensis*). The inventory showed that although the tree composition in the plots changed over time, no large variations in density or basal area occurred during the study period (Table 1; Fig 3).

## Data analyses

**Competition index and search radius.** We calculated two CIs for each sampled tree using the Hegyi index with DBH (Eq 1) and crown volume (Eq 2) as variables. The Hegyi index is used to estimate the level of competition by weighting the contribution of each competitor based on size and distance as follows [26]:

$$\text{CI} = \Sigma(Dj/Di) \times (1/Lij + 1) \qquad (1)$$

$$\text{CI} = \Sigma(CVj/CVi) \times (1/Lij + 1) \qquad (2)$$

Where D is the DBH of competitor tree j and target tree i. Lij is the distance between the competitor and target tree. CV is the crown volume of competitor tree j and target tree i. This study assumes that the tree canopy is conical. The formula for crown volume is $CV = 1/3\pi$

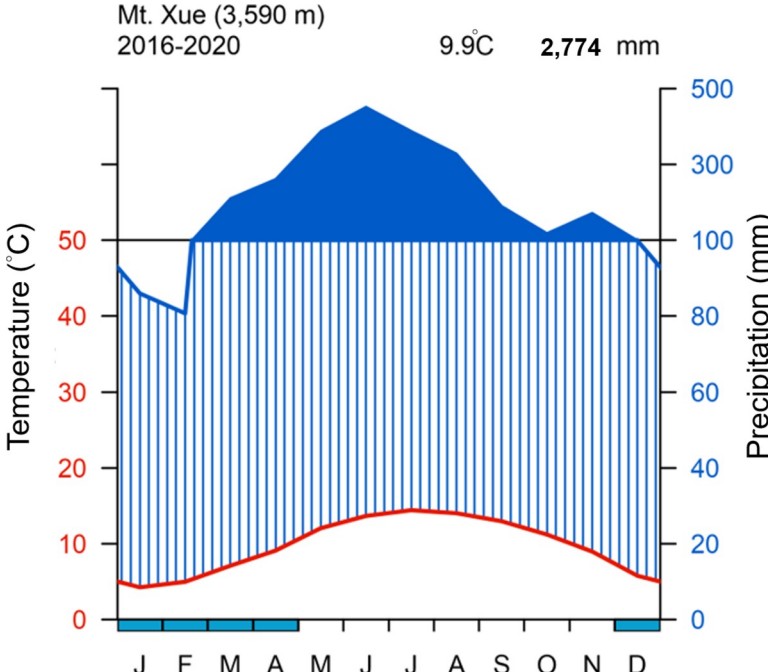

**Fig 2. Climate diagram in the study area on Mt. Xue in Taiwan.** Plots were drawn in R 3.6.3(Cran: climatol, Cairo); Meteorological data was from CWB observation data inquire system (https://e-service.cwb.gov.tw/HistoryDataQuery/index.jsp). This original content was made in this study under the CC BY 4.0 license.

$\times$ (CR)$^2$ $\times$ (TH−CL), where CR is the crown radius, TH is the tree height, and CL is the clear length.

The Hegyi index was calculated for all trees within a certain radius [27] of the target tree. When DBH was used for the calculation, we referenced two equations developed using a fixed search radius (D1) [26] and a variable search radius (D2) [28]. The fixed search radius divides the DBH of surviving trees into three levels. A DBH < 8 cm indicated a search radius of 3.04 m (0.005 ha), a DBH of 8–16 cm indicated a search radius of 3.99 m (0.01 ha), and a DBH > 16 cm indicated a search radius of 5.65 m (0.02 ha). Different competitor's sizes were used to capture local processes affecting large and small trees. Because demographic processes

**Table 1. Species composition in seven black forest plots on Mt. Xue in Taiwan in 2008.**

| Dominant overstory species | A1 | A2 | A3 | A4 | A5 | A6 | A7 |
|---|---|---|---|---|---|---|---|
| *Abies kawakamii* | 79 (35.59) | 119 (48.57) | 116 (85.93) | 217 (99.09) | 58 (28.85) | 116 (92.8) | 101 (73.19) |
| *Tsuga chinensis* var. *formosana* | 142 (63.96) | 5 (2.04) | 1 (0.74) | 0 (0) | 0 (0) | 0 (0) | 0 (0) |
| *Juniperus formosana* | 1 (0.45) | 0 (0) | 0 (0) | 0 (0) | 0 (0) | 0 (0) | 0 (0) |
| *Juniperus squamata* var. *morrisonicola* | 0 (0) | 4 (1.63) | 0 (0) | 1 (0.46) | 0 (0) | 0 (0) | 35 (25.36) |
| *Sorbus randaiensis* | 0 (0) | 15 (6.12) | 6 (4.44) | 1 (0.46) | 143 (71.14) | 9 (7.2) | 2 (1.45) |
| *Rhododendron pseudochrysanthum* | 0 (0) | 102 (41.63) | 12 (8.89) | 0 (0) | 0 (0) | 0 (0) | 0 (0) |
| **Total individuals** | 222 | 245 | 135 | 219 | 201 | 125 | 138 |
| **Dead individuals (unconfirmed/*A. kawakamii*)** | 14/35 | 27/6 | 18/37 | 42/41 | 16/16 | 39/18 | 22/1 |
| **Number of understory species** | 26 | 25 | 24 | 29 | 41 | 21 | 46 |
| **Dominant understory type** | bamboo | bamboo | bamboo | moss | moss | moss | moss |

Data are presented as the number of individuals (percentage of total) unless otherwise stated.

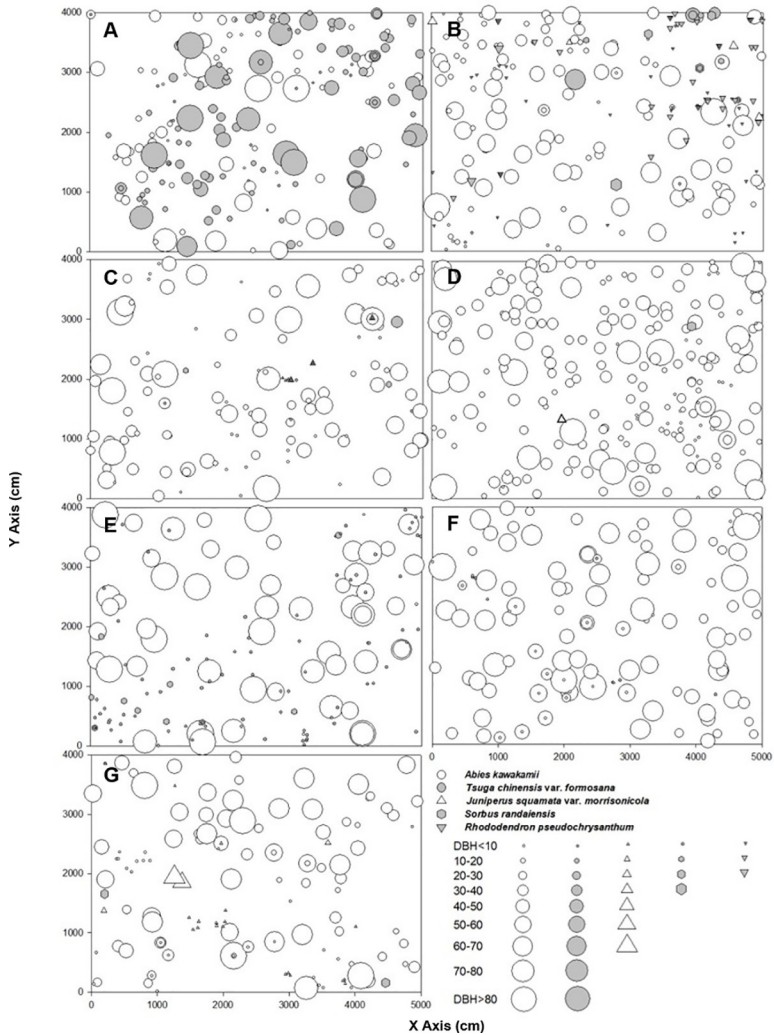

**Fig 3. Individual spatial patterns of each overstory species in the study area on Mt. Xue.** A was in A1, B was in A2, C was in A3, D was in A4, E was in A5, F was in A6, G was in A7. The understory of A–C was composed of *Yushania niitakayamensis*; The understory of D–G was composed of moss.

operate across ranges rather than discrete distances, we evaluated the aggregate spatial patterns throughout the 6 m radius of the neighborhood. Referring to the results of Wang et al. [28], if the sum of the DBH of the competitor tree (CT) and the target tree (TT) divided by four is greater than the distance between the two trees, these individuals are considered competitors.

In addition, when the competition index is calculated by crown volume (T1), if the distance between the TT and CT is less than $1/\sqrt{3}$ TH of the CT, the individuals are considered competitors. In other words, the root of TT is the original point which Shot at an angle of 60˚, and the trees that meet it are all competitor trees [29, 30]. Because edge bias will underestimate the results of the competition [31], we arrange every plot into nine blocks [32], but only calculate the CI of the TT in the central block to reduce the boundary influence deviation [28].

**Analysis of CI computed by different CT and TT.** In this study, we measured SSAs, such as DBH, TH, CL, and CV, and (1) explored the relationships among the CI and SSAs; (2) compared CIs between inter- and intraspecies, health status (living and dead individuals); and (3) evaluated the relationship between the DBH and competition index. First, we used Pearson's

correlation analyses to examine the relationships between the CI and other variables. Second, we compared CIs among different species and dead and living trees, testing the significance in each plot using the Kruskal-Wallis test. Finally, we constructed a scatter plot which charting the results of the individual's DBH and CI, then fitting a regression function and Bonferroni correction to determine the *p*-value of the resulting pattern.

## Results and discussion

### Relationships between the competition index and stand structural attributes

This study used three search radius methods to sort out different competitive pressures (Tables 2 and 3). When inter-and intraspecies competition occurs, the D2 search radius had the highest number of competitive trees, followed by T1, and finally D1. In addition, the DBH variability of the competitive tree with a D2 search radius is more stable than that of D1, even though the average DBH of the competitive tree in D2 is higher than in D1. Wang et al.'s [28] study on growth competition in mixed plantations indicated that the D2 search radius got the better effect when CI was calculated using DBH. The larger search radius of D2 includes more competitor trees, resulting in a more accurate calculation of the number of competitor trees that affect the TT [28, 33].

Although the number of competitor trees was slightly higher in the T1 search radius than in D1, the number of CT is still quite limited. High-strength environmental disturbances at high

**Table 2. Intra- and interspecific competition of *Abies kawakamii* population on Mt. Xue.**

| Type of search radius | | TT | | Intraspecies CT | | Interspecies CT | |
|---|---|---|---|---|---|---|---|
| | Plot | No. | DBH (cm) | No. | DBH (cm) | No. | DBH (cm) |
| D1 | A1 | 33 | 22.2 ± 19.36 | 6 ± 3 | 16.42 ± 5.04 | 7 ± 5 | 31.43 ± 27.98 |
| | A2 | 28 | 30.98 ± 18.17 | 4 ± 2 | 29.36 ± 14.16 | 3 ± 2 | 10.70 ± 7.89 |
| | A3 | 45 | 36.68 ± 24.95 | 8 ± 6 | 24.29 ± 15.10 | 3 ± 1 | 18.71 ± 16.12 |
| | A4 | 93 | 28.67 ± 20.78 | 13 ± 5 | 26.22 ± 19.93 | 1 ± 0 | 14.97 ± 11.38 |
| | A5 | 20 | 61.46 ± 17.07 | 3 ± 2 | 63.44 ± 9.65 | 3 ± 2 | 12.78 ± 16.80 |
| | A6 | 34 | 41.74 ± 18.71 | 4 ± 2 | 46.27 ± 11.21 | 3 ± 2 | 24.60 ± 12.19 |
| | A7 | 25 | 33.06 ± 23.23 | 3 ± 2 | 34.22 ± 14.19 | 2 ± 2 | 21.85 ± 16.34 |
| D2 | A1 | 114 | 22.2 ± 19.36 | 30 ± 32 | 32.86 ± 4.42 | 73 ± 32 | 50.17 ± 10.14 |
| | A2 | 61 | 30.98 ± 18.17 | 35 ± 17 | 38.44 ± 5.16 | 41 ± 37 | 26.66 ± 8.87 |
| | A3 | 151 | 36.68 ± 24.95 | 99 ± 70 | 56.76 ± 5.95 | 13 ± 13 | 38.7 ± 13.58 |
| | A4 | 256 | 28.67 ± 20.78 | 103 ± 81 | 47.43 ± 5.43 | 16 ± 14 | 64.84 ± 22.36 |
| | A5 | 74 | 61.46 ± 17.07 | 112 ± 12 | 66.08 ± 0.86 | 77 ± 40 | 12.21 ± 3.42 |
| | A6 | 134 | 41.74 ± 18.71 | 97 ± 44 | 50.60 ± 2.50 | 20 ± 13 | 47.71 ± 13.13 |
| | A7 | 101 | 33.06 ± 23.23 | 53 ± 33 | 49.59 ± 5.04 | 21 ± 18 | 41.91 ± 11.84 |
| H1 | A1 | 114 | 22.2 ± 19.36 | 18 ± 4 | 32.58 ± 2.19 | 22 ± 7 | 52.42 ± 9.33 |
| | A2 | 61 | 30.98 ± 18.17 | 21 ± 5 | 40.80 ± 5.18 | 2 ± 2 | 17.93 ± 13.6 |
| | A3 | 151 | 36.68 ± 24.95 | 51 ± 5 | 52.64 ± 2.86 | 2 ± 1 | 18.29 ± 6.86 |
| | A4 | 256 | 28.67 ± 20.78 | 31 ± 5 | 43.41 ± 4.44 | 1 ± 0 | 23.18 ± 8.83 |
| | A5 | 74 | 61.46 ± 17.07 | 16 ± 4 | 70.08 ± 4.82 | 2 ± 2 | 5.39 ± 1.68 |
| | A6 | 134 | 41.74 ± 18.71 | 26 ± 4 | 50.68 ± 2.43 | 1 ± 1 | 21.6 ± 1.00 |
| | A7 | 101 | 33.06 ± 23.23 | 10 ± 3 | 14.60 ± 12.79 | 1 ± 1 | 65.04 ± 0.91 |

TT, target tees; CT, competitor trees; DBH, diameter at breast height; D1, CI is calculated by DBH and fixed distance; D2, CI is calculated by DBH and variable distance; T1, CI is calculated by crown volume and variable distance.

**Table 3. Dead and live competition of *Abies kawakamii* population on Mt. Xue.**

| Type of search radius | Plot | Dead TT | | Live TT | | CTs of dead | | CTs of live | |
|---|---|---|---|---|---|---|---|---|---|
| | | No. | DBH (cm) | No. | DBH (cm) | No. | DBH (cm) | No. | DBH (cm) |
| D1 | A1 | 19 | 15.56 ± 11.41 | 14 | 19.97 ± 11.75 | 3 ± 2 | 21.16 ± 23.56 | 2 ± 1 | 21.85 ± 24.11 |
| | A2 | 1 | 17.20 | 27 | 38.70 ± 16.17 | 1 | 36.62 ± 22.19 | 2 ± 1 | 23.71 ± 18.93 |
| | A3 | 12 | 39.93 ± 44.80 | 33 | 32.00 ± 25.30 | 4 ± 3 | 30.47 ± 26.73 | 3 ± 2 | 24.60 ± 19.19 |
| | A4 | 23 | 26.15 ± 25.15 | 70 | 26.24 ± 18.12 | 2 ± 1 | 24.79 ± 18.61 | 2 ± 1 | 15.47 ± 9.19 |
| | A5 | 4 | 43.80 ± 10.77 | 16 | 71.32 ± 14.34 | 1 ± 0 | 38.63 ± 33.93 | 2 ± 1 | 36.78 ± 30.30 |
| | A6 | 3 | 27.63 ± 7.39 | 31 | 45.11 ± 17.40 | 1 ± 0 | 37.66 ± 19.51 | 3 ± 2 | 37.16 ± 21.44 |
| | A7 | - | - | 25 | 36.24 ± 20.70 | - | - | 3 ± 1 | 37.50 ± 11.51 |
| D2 | A1 | 35 | 13.74 ± 10.22 | 79 | 26.08 ± 15.24 | 85 ± 73 | 46.67 ± 7.03 | 110 ± 119 | 44.17 ± 7.48 |
| | A2 | 9 | 31.3 ± 20.10 | 52 | 30.97 ± 18.20 | 101 ± 72 | 34.62 ± 6.70 | 97 ± 66 | 35.07 ± 5.82 |
| | A3 | 34 | 32.7 9± 32.41 | 117 | 37.92 ± 22.43 | 110 ± 134 | 55.86 ± 7.18 | 114 ± 68 | 51.11 ± 6.16 |
| | A4 | 41 | 26.87 ± 30.54 | 215 | 28.96 ± 18.81 | 127 ± 168 | 51.16 ± 7.08 | 119 ± 77 | 48.57 ± 6.07 |
| | A5 | 16 | 46.27 ± 13.84 | 58 | 65.13 ± 15.77 | 140 ± 48 | 46.74 ± 4.36 | 200 ± 68 | 44.90 ± 5.61 |
| | A6 | 18 | 27.41 ± 8.87 | 116 | 43.47 ± 18.86 | 78 ± 21 | 51.16 ± 2.53 | 68 ± 18 | 51.94 ± 2.09 |
| | A7 | 1 | 73.50 | 100 | 32.66 ± 22.98 | 153 | 37.77 ± 27.12 | 73 ± 51 | 47.17 ± 6.07 |

TT, target tees; CT, competitor trees; DBH, diameter at breast height; D1, CI is calculated by DBH and fixed distance; D2, CI is calculated by DBH and variable distance; T1, CI is calculated by crown volume and variable distance. Plot A7 had no dead *Abies kawakamii*.

altitudes (such as strong winds, typhoons, or heavy rainfall) can damage trees, making it difficult to measure CV and identify species. Therefore, the calculation of dead individuals of *A. kawakamii* is often biased, and T1 cannot be adopted.

The results of calculating the competitive pressure on the health status of the TT show that the number of competitor trees in the D2 search radius was greater than in D1 (Table 3). Make a small conclusion; our study also shows that the search radius D2 are obviously more reliable than the other two search methods. Therefore, the subsequent analysis of the competition index and different types of competition sources will focus on the D2 search radius. In addition, according to different types of competitive pressure, Tables 1 and 2 show that the average DBH of most intraspecies competitor trees was greater than those of interspecies competitors, and the average DBH of dead trees was much greater than those of the target trees.

Most SSAs were highly correlated (DBH, TH, CL, and CV) (Tables 4 and 5). Comparing between *A. kawakamii* plots with different dominant understories, the CI was higher when the understory was composed of moss than when it was composed of *Y. niitakayamensis*. Furthermore, in plots where the understory mainly was moss, most SSAs were negatively correlated

**Table 4. Correlations between *Abies kawakamii* stand structural attributes in plots with a *Yushania niitakayamensis*-dominant understory on Mt. Xue.**

| Stand structural attributes (SSAs) | DBH | TH | CL | CV | CI |
|---|---|---|---|---|---|
| Diameter at breast height (DBH) | | 0.35 | 0.36 | 0.20 | 0.05 |
| Tree height (TH) | **0.59** | | 0.53 | 0.91 | 0.00 |
| Clear length (CL) | **0.60** | **0.73** | | 0.25 | 0.00 |
| Crown volume (CV) | **0.45** | **0.96** | **0.50** | | 0.00 |
| Hegyi competition index with D2 (CI-D2) | **0.23** | 0.04 | 0.06 | -0.01 | |

Bold font indicates significance at $P < 0.05$. Bottom left is the correlation coefficient; top right is the coefficient of determination.

**Table 5. Correlations between *Abies kawakamii* stand structural attributes in plots with a moss-dominant understory on Mt. Xue.**

| stand structural attributes (SSAs) | DBH | TH | CL | CV | CI |
|---|---|---|---|---|---|
| Diameter at breast height (DBH) | | 0.59 | 0.52 | 0.57 | 0.07 |
| Tree height (TH) | **0.77** | | 0.84 | 0.76 | 0.06 |
| Clear length (CL) | **0.72** | **0.92** | | 0.39 | 0.04 |
| Crown volume (CV) | **0.75** | **0.87** | **0.62** | | 0.08 |
| Hegyi competition index with D2(CI-D2) | −0.26 | −0.25 | −0.21 | −0.28 | |

Bold font indicates significance at $P < 0.05$. Bottom left is the correlation coefficient; top right is the coefficient of determination.

with CI ($P < 0.05$). Conversely, in plots where bamboo-dominated the understory, most SSAs were not significantly correlated with CI, and the DBH was positively correlated with CI ($P < 0.05$). The creeping rhizomes of *Y. niitakayamensis* rapidly cover space, gaining biomass after canopy gaps are produced and prohibiting *A. kawakamii* seedling and sapling growth [17]. Thus, in areas with *Y. niitakayamensis*-dominant understory, *A. kawakamii* competes with the bamboo as well as other woody species. Therefore, these SSAs were correlated, and the correlation with the moss understory was higher than that with *Y. niitakayamensis*.

These results are in line with many previous studies [8, 10, 34, 35]. Competition has a negative effect on the characteristics of the target tree. Katharina et al. [10] suggest that the resource acquisition capacity of trees is negatively affected by competition. Accumulation of crown volume is a concern with productivity that determines leaf area and impacts light interception and the microclimate of the canopy. Furthermore, trees change their morphological characteristics under competition stress to adapt to the limited resource availability in the environment [36]. We also found interspecific competition between bamboo and woody plants, which could be attributed to the biological characteristics of bamboo, which quickly occupies open space and consumes nutrient resources in its early stages [17].

We combined the CI and individual distribution plots to determine the distribution of competition (Fig 4). High competition occurred where there were gatherings of live and dead individuals. To confirm these results, we separated the dead and living individuals to compare their CI independently.

## Comparing inter- and intraspecific competition indices of *A. kawakamii* individuals

The intraspecific CI was significantly higher than the interspecific CI in all plots, except A1 and A2, where the interspecific CI was significantly higher (Table 6). The A1 and A2 plots had higher proportions of other species (A1, *T. chinensis* var. *formosana* 142 (63.96%); A2, *R. pseudochrysanthum* 102 (41.63%); Table 1) and more interspecific competition against target trees, whereas plots A3–A7 had greater numbers of intraspecific competing trees (Table 2). In addition, *A. kawakamii* and *T. chinensis* var. *formosana* are both shade-intolerant trees in the Pinaceae family that require gaps to regenerate [37–40]. The overlap of ecological niches leads to the coexistence of the two species being maintained in the form of competition [17, 20]. A similar situation also occurs within the *A. kawakamii* population. The spatial distribution of *A. kawakamii* in Fig 3 shows that the distribution of saplings (DBH < 20 cm) is mostly clustered. Therefore, the intraspecific CI in most plots was relatively high. Differences in the ground cover composition may also intensify competition for resources, especially in alpine environments [17]. However, it is difficult to quantify the suppressing influence of bamboo on *A. kawakamii*.

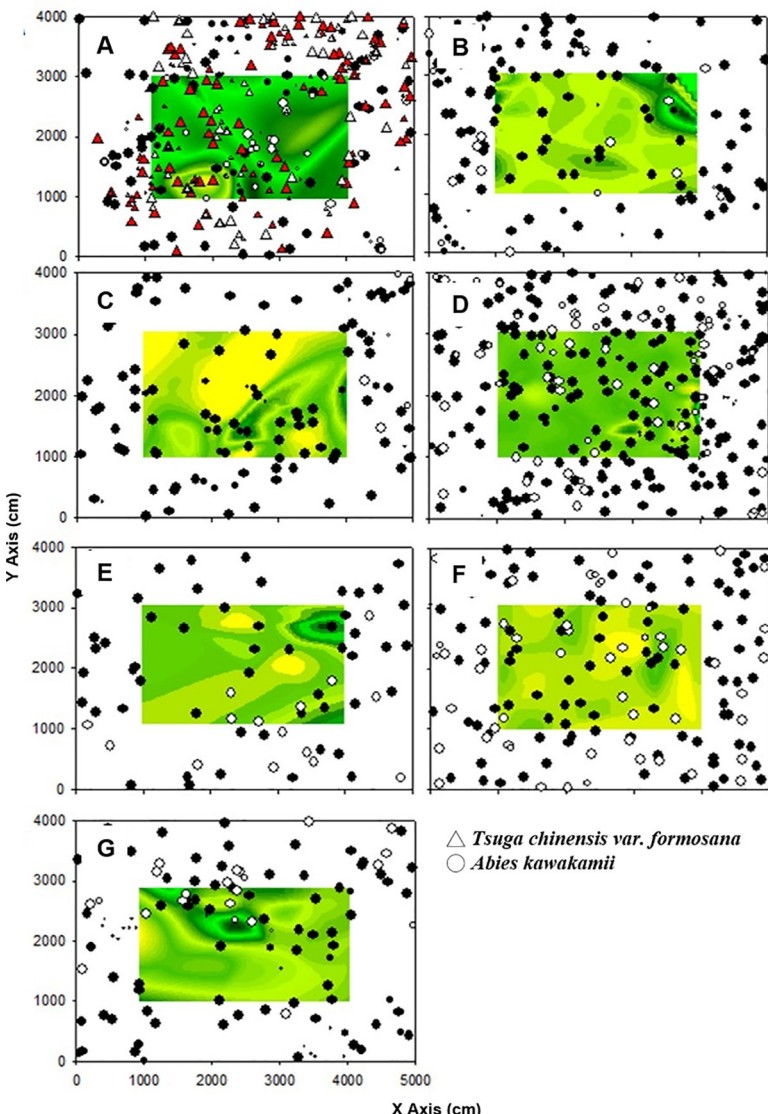

**Fig 4. Spatial patterns with competition index curves of individual trees in the study plots on Mt. Xue.** Solid circles indicate living individuals, and empty circles indicate dead individuals. The background gradient graphic shows the competition index from high (white) to low (black) values. A was in A1, B was in A2, C was in A3, D was in A4, E was in A5, F was in A6, G was in A7. The understories of A–C were dominated by *Yushania niitakayamensis*; those of D–G were dominated by moss.

## Comparing competition indices among living and dead *A. kawakamii* individuals

We assumed that the competition index would be higher among dead individuals than among living individuals [41, 42], so we divided individuals into groups to test this hypothesis. Table 7 shows that the CIs were higher among dead individuals of *A. kawakamii* than among living individuals. However, one plot with a *Y. niitakayamensis*-dominated understory (A2) yielded similar CIs among dead and living individuals.

Generally, our results showed that dead individuals exhibited a higher CI compared to living individuals, indicating that dead individuals experience more competitive pressure. This

**Table 6. Intra- and interspecific competition indices (CI) of target trees (TT; *Abies kawakamii*) on Mt. Xue.**

| Plot | Mean of CI | | No. of TT | $\chi^2$ | *P*-value |
|------|------------|---|-----------|----------|-----------|
| | Intraspecific | Interspecific | | | |
| A1 | 0.07 ± 0.10 | **0.14 ± 0.12** | 114 | 53.81 | < 0.05 |
| A2 | 0.05 ± 0.05 | **0.24 ± 0.40** | 61 | 8.24 | < 0.05 |
| A3 | **0.18 ± 0.18** | 0.01 ± 0.02 | 151 | 137.25 | < 0.05 |
| A4 | **0.20 ± 0.13** | 0.03 ± 0.04 | 256 | 363.79 | < 0.05 |
| A5 | **0.12 ± 0.18** | 0.01 ± 0.00 | 74 | 107.25 | < 0.05 |
| A6 | **0.21 ± 0.30** | 0.00 ± 0.00 | 134 | 153.94 | < 0.05 |
| A7 | **0.15 ± 0.17** | 0.04 ± 0.06 | 101 | 88.48 | < 0.05 |

Bold font indicates a significantly higher CI within the same plot.

result is consistent with the report of Tang et al. [43]. The present study shows that competition significantly increased tree mortality. As discussed above, the regeneration of the *A. kawakamii* population is related to gaps, and many seedlings and saplings are often grouped together in these gaps. In addition to competition for nutrients, the environmental screening intensity is high, limited by the existence, size, and location of gaps. Factors such as understory composition and growth space can reduce the survival rate of seedlings [38, 44]. Most studies on the population structure of *A. kawakamii* report high seedling and sapling mortality rates [17–19, 37, 38, 45]. The self-thinning phenomenon of *A. kawakamii* during the sapling period is widely reported; thus, we evaluated the relationship between the DBH and competition index. However, in addition to the biological characteristics of *A. kawakamii*, our survey results suggest that the competitive pressure on dead individuals may also be affected by the type of understory and the micro-environments of gaps.

## The relationship between DBH and competition index

We constructed a scatter plot to examine the relationship between the competition index and DBH. Three distinct patterns between CI and DBH were observed (Fig 5). In plots A1 and A2, a positive linear regression was detected. In plots A3, A4, and A7, the CI along the different DBH values showed a descending power formula. No significant patterns with CI and DBH were observed in plots A5 and A6 ($P > 0.05$). Based on the previous results [46, 47], we predicted that the DBH of *A. kawakamii* would negatively relate to the CI; however, only a few areas showed similar configurations. Except for the well-suited plots with descending power

**Table 7. Competition indices (CI) of dead and live target trees (TT; *Abies kawakamii*) on Mt. Xue.**

| Plot | Mean of CI | | No. of TT | | $\chi^2$ | *P*-value |
|------|------------|---|-----------|---|----------|-----------|
| | Dead | Live | Dead | Live | | |
| A1 | **0.31 ± 0.18** | 0.17 ± 0.14 | 35 | 79 | 25.53 | < 0.05 |
| A2 | 0.13 ± 0.16 | 0.33 ± 0.43 | 9 | 52 | 2.03 | 0.16 |
| A3 | **0.25 ± 0.18** | 0.18 ± 0.18 | 34 | 117 | 7.39 | < 0.05 |
| A4 | **0.26 ± 0.14** | 0.22 ± 0.15 | 41 | 215 | 4.62 | < 0.05 |
| A5 | **0.13 ± 0.12** | 0.13 ± 0.19 | 16 | 58 | 10.07 | < 0.05 |
| A6 | **0.24 ± 0.26** | 0.22 ± 0.30 | 18 | 116 | 7.44 | < 0.05 |
| A7 | 0.06 | 0.19 ± 0.21 | 1 | 100 | - | - |

TT, target trees. Bold font indicates a significantly higher CI within the same plot. Plot A7 only had one dead individual; thus, no test was performed.

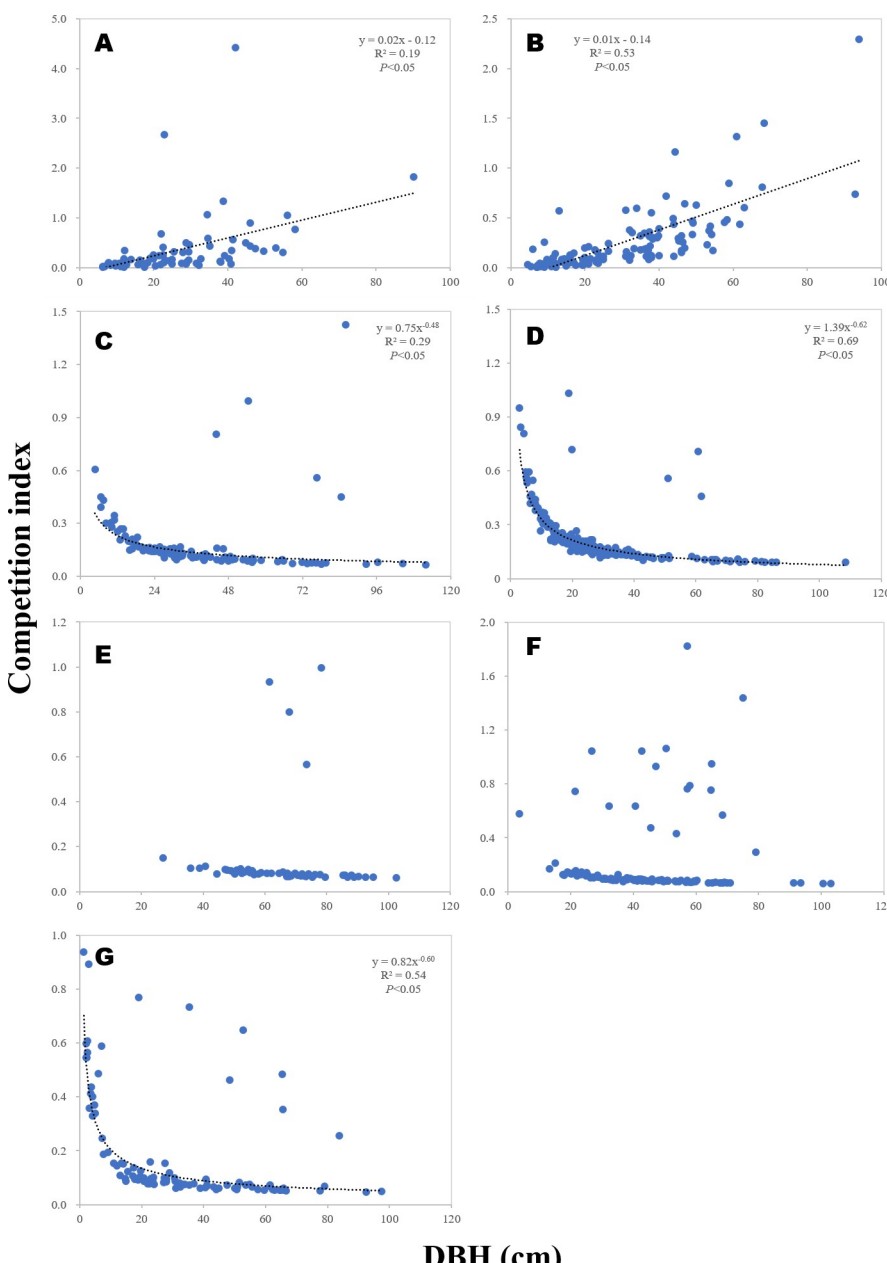

**Fig 5. Scatter plot of the DBH and competition index of *Abies kawakamii* and *Tsuga chinensis* var. *formosana* samples on Mt. Xue.** Different letters indicate different samples. A was in A1, B was in A2, C was in A3, D was in A4, E was in A5, F was in A6, G was in A7. The understories of A–C were dominated by *Yushania niitakayamensis*; those of D–G were dominated by moss.

functions, most plots have relatively few seedlings and saplings, especially A5 and A6. The low numbers of seedlings and saplings in plots A5 and A6 could be due to past disturbances. Individuals are disturbed seriously that could not be identified or collapsed, show a linear downward trend in the second half of the power function.

In addition, the A1 and A2 plots are located at the intersection of *A. kawakamii* and *T. chinensis* var. *formosana* [17], and competition from other species (including bamboo) is

relatively high. The structures of *A. kawakamii* plots with bamboo-dominated understories are comparable to those reported by Taylor and Qin [48]. The regeneration process of *A. faxoniana* is similar to the fluctuation of the growth cycle with *Bashania fangiana*. Therefore, when the target tree DBH is larger, the number of competing trees and the CI value are correspondingly larger. However, few of the sample coefficients of determination were low ($R^2 < 0.50$). As the DBH increases, the competition index decreases, consistent with patterns observed by many other researchers [8, 10, 35, 49].

Charles et al. [49] suggested that a large DBH indicated that the survivors of various growth periods had maintained sufficient distances from adjacent survivors to avoid competition for space and resources. Furthermore, because seedlings and saplings grow as soon as gaps appear in the canopy, their distributions seem clumped and dense under these gaps. Therefore, individuals with a small DBH demonstrate a correspondingly high competition index. Yu et al. [7] indicated that vegetation comprised of simple species would be more influenced by intraspecies competition than by interspecies competition, particularly in the case of seedlings and saplings that are sensitive to the environment and resources. Therefore, small individuals exhibit high CIs.

In this study area, the samples are frequently disturbed by fire, causing an increase in the CI among *Pinus sylvestris* saplings [7]. Such sources of disruption or other unspecified variables can obfuscate the relationship between dead seedlings and saplings and the CI in small DBH samples. This could explain why A1 of the *A. kawakamii* population exhibited a discontinuous pattern, whereas A1 of the *T. chinensis* var. *formosana* population exhibited a smooth descending power formula pattern. The altitude for A1 might have been too low for the *A. kawakamii* population, yielding few dead individuals. In summary, *A. kawakamii* living individuals maintain distance from other individuals over time via competition.

## Conclusions

In this study, we measured DBH and distance between individuals to quantify the level of competitive interaction. We used various data of CT and TT to compare results to determine the relationship between populations in the subalpine region in Taiwan. We fix the hypothesis into these results: (1) CI based on variable distance would search more competitor trees than CI based on fixed distance. (2) When bamboo dominant understory, CI is positive corelated with most SSAs; When moss dominant understory, CI is negative corelated with SSAs. (3) Individuals of Taiwan fir would be sustained more competitive stress from intraspecies than interspecies. (4) Dead individuals would be endured more competitive stress than living individuals. (5) In this study, we find three patterns between DBH and CI, it's may cause by understory types and disturbance from environment. Although we qualified the level of competitive interaction, varying radii may yield different outcomes. To ensure that this study was unbiased, an additional function should be used to verify that the analyses were positive and unbiased.

## Supporting information

**S1 File. Competition indice of *Abies kawakamii* individuals in this research.** D/L means the health status of individuals. D: dead individuals; L: live individuals. TH: tree height, CL: clear length, CV: crown column.
(XLSX)

## Acknowledgments

We are grateful to the Shei-Pa National Park of Taiwan for their assistance with the fieldwork. We are greatful to Enago (www.enago.tw) edited this manuscript.

## Author Contributions

**Conceptualization:** Min-Chun Liao.

**Formal analysis:** Wei Wang.

**Investigation:** Min-Chun Liao.

**Methodology:** Wei Wang, Min-Chun Liao.

**Project administration:** Wei Wang, Min-Chun Liao.

**Resources:** Min-Chun Liao.

**Supervision:** Hsy-Yu Tzeng.

**Writing – original draft:** Wei Wang, Min-Chun Liao.

**Writing – review & editing:** Min-Chun Liao, Hsy-Yu Tzeng.

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
