## [Decision Letter · Decision Letter 0]

25 Feb 2021

PONE-D-20-38825

Intraspecific competition in Abies kawakamii forest at subtropical high mountain in Taiwan.

PLOS ONE

Dear Dr. Tzeng,

Thank you for submitting your manuscript to PLOS ONE. After careful consideration, we feel that it has merit but does not fully meet PLOS ONE’s publication criteria as it currently stands. Therefore, we invite you to submit a revised version of the manuscript that addresses the points raised during the review process.

Reviewer #1 raised important objections against the methodology, which need to be considered.

Moreover, the manuscript lacks a proper discussion. Even when joining the chapters Results and Discussion is allowed by authors' guidelines, the joint chapter cannot be just descriptive and must set the results into a meaningful biological context. I suggest splitting the chapter into separate parts to ensure this.

We look forward to receiving your revised manuscript.

Kind regards,

Dusan Gomory

Academic Editor

PLOS ONE

Journal Requirements:

"The funders had no role in study design, data collection and analysis, decision to

publish, or preparation of the manuscript."

3. We note that Figure 1 in your submission contains map images which may be copyrighted. All PLOS content is published under the Creative Commons Attribution License (CC BY 4.0), which means that the manuscript, images, and Supporting Information files will be freely available online, and any third party is permitted to access, download, copy, distribute, and use these materials in any way, even commercially, with proper attribution. For these reasons, we cannot publish previously copyrighted maps or satellite images created using proprietary data, such as Google software (Google Maps, Street View, and Earth). For more information, see our copyright guidelines: http://journals.plos.org/plosone/s/licenses-and-copyright.

(1) You may seek permission from the original copyright holder of Figure 1 to publish the content specifically under the CC BY 4.0 license. 

Reviewers' comments:

Reviewer's Responses to Questions

**Comments to the Author**

1. Is the manuscript technically sound, and do the data support the conclusions?

Reviewer #1: No

Reviewer #2: Yes

2. Has the statistical analysis been performed appropriately and rigorously? 

Reviewer #1: No

Reviewer #2: Yes

3. Have the authors made all data underlying the findings in their manuscript fully available?

Reviewer #1: No

Reviewer #2: Yes

4. Is the manuscript presented in an intelligible fashion and written in standard English?

Reviewer #1: No

Reviewer #2: Yes

5. Review Comments to the Author

Reviewer #1: General comments

Authors quantified the intraspecific competition impact to Abies kawakamii trees in the mixed forests at the sub-tropical region of Taiwan. Authors used the Hegyi’s competition index (CI) formula, which is most commonly used in forest modeling, to quantify competition impact. Authors used DBH and inter-tree distance to compute the index, disregarding other tree attributes, such as crown dimensions and tree heights. Authors used one-time measurement data acquired from seven sample plots. Estimated CI was then used to correlate with various tree attributes, such as DBH, height, crown dimensions, which authors defined them as ecological attributes. Authors presented the significant correlations of CI with each of these attributes, which is quite obvious. Authors also presented the results of higher CI for dead trees than that for living trees, indicating that such dead trees may be due higher competition impact.

This study is very simple, but classical-like. Manuscript thus does not provide any new knowledge to the audience, because of applying the classical methods of computing CI from smaller data set acquired from only seven sample plots. Authors used the fixed search radius approach for finding the competitor around the subject tree, which is less robust approach and it should have been used with some improvements, such as variable radius approach. The classical approach of identifying the potential competitors is less precise than the other more robust approaches, such as horizontal search angle and vertical search angle (inverted cone) approaches. Most of the forest modeling studies conducted in recent years have used these approaches. Showing only the correlations of CI with other tree attributes cannot be innovative, as previous studies have shown such relationships. It would be more interesting and useful when CI computed with search angle (horizontal or vertical) approach and used as one of the predictors of tree growth models (e.g. radial growth models) and examining the impact of competition to growth. Furthermore, authors only computed DBH-based CI, which would be less precise than the CI computed using tree crown dimensions. Thus, CI should be computed based on the various tree attributes separately, such as crown dimensions (e.g. crown area, crown volume), height and DBH, and compared the contributions to the growth variations or any other tree attributes variations. In-depth analysis and comparison of CIs computed using different search angle approaches with different tree attributes should be done and the best CI and search angle approach identified. However, authors did not attempt do so, and growth analysis is also not possible with authors’ data set, which was measured only once. Based on the methodological weakness of the study as main problem, and other problems, such as fewer sample plots, and poorly formulated texts including incomplete coverage of relevant literature studies and citations, lack of data summary statistics, etc., this manuscript does not meet the minimum standard of PONE. Thus, authors are suggested to find other outlet for their work.

Reviewer #2: General comment

Minor revision is required.

Specific comments regarding what needs to be revised:

Lines 73-74. The Introduction is generally OK, but no strong explanation has been provided on why the study was conducted in the first place. What is its importance/contribution to forest science? With that in mind, I suggest that you add one or two sentences before the last sentence in the Introduction, in which you could mention, for instance, that a number of studies focused on relationships of structural indices and structural attributes have already been conducted (here you may cite, for instance, the following articles:

https://doi.org/10.3390/f11010004

or

https://doi.org/10.1016/j.baae.2018.02.007

Then, you may emphasize the novelty of your research because you investigated the relationships of competition indices and structural (ecological) attributes.

Lines 102-104. Rephrase or delete the senstence "The seven plots ..." as it makes no sense. Various stand characteristics are not sample size, just different variables, so they do not minimize unreliable inferences. Also, seven plots is rather small sample size, but I accept their importance due to detailed measurements.

Line 107. What was the study period? Please specify, or otherwise delete.

Line 108. Add one sentence that will describe the number and the size of plots (in m2 or ha).

Lines 138. In the section Results, whenever you describe any correlation, please add corresponding R2 values, not only descriptive text.

Line 145. Add in brackets which ecological attributes?

Lines 192-193. You mention here that dead individuals exhibited a higher competitive index compared to the living individuals. But is this statistically significant? If so, provide computed p-value of the applied statistical test.

Line 226. Replace "significant individuals" with "large individuals"

In the text below Figure 4, replace "Scatter plot" with "Box-plot" as these are obviously box-plots.

Discussion and Conclusions are properly written.

6. PLOS authors have the option to publish the peer review history of their article (what does this mean?). If published, this will include your full peer review and any attached files.

Reviewer #1: No

Reviewer #2: No

---

## [Author Response · Author response to Decision Letter 0]

23 May 2021

Reviewer #1

Q1. Authors used DBH and inter-tree distance to compute the index, disregarding other tree attributes, such as crown dimensions and tree heights.

Reply to Q1. We add competitive index (CI) count by crown volume (CV) (Line 136-140, 160-163), but we suffer dilemma that we still use CI computed by DBH (Line 194-201).

Q2. Authors used the fixed search radius approach for finding the competitor around the subject tree, which is less robust approach and it should have been used with some improvements, such as variable radius approach.

Reply to Q2. Thank your comment, we reexamine our data and count CI by variable radius(D2), and we got better result in this article (Line154-159, 185-192, 205-212), so that we use CI-D2 to talk over following chapters. 

Q3. Based on the methodological weakness of the study as main problem, and other problems, such as fewer sample plots, and poorly formulated texts including incomplete coverage of relevant literature studies and citations, lack of data summary statistics.

Reply to Q3. We modified our description about method & analysis (Line 130-180). We also add more relevant literature studies and citations (References 6, 19, 20, 27-30, 32-34, 38-41, 43, 45, 46). Moreover, we list more detail information about data summary statistics (Table1-7).

Reviewer #2

Q1. The Introduction is generally OK, but no strong explanation has been provided on why the study was conducted in the first place.

Reply to Q1. Thank your comment, we follow your recommendation to add a description of this manuscript's importance (Line 74-79).

Q2. What was the study period? Please specify, or otherwise delete. Add one sentence that will describe the number and the size of plots (in m2 or ha).

Reply to Q2. We add the description of study period and sample size in Line 116 & 123.

Q3. In the section Results, whenever you describe any correlation, please add corresponding R2 values, not only descriptive text. You mention here that dead individuals exhibited a higher competitive index compared to the living individuals. But is this statistically significant? If so, provide computed p-value of the applied statistical test.

Reply to Q3. We modified all table which list P value & R2 (Table 4-7). Even though we compute competitive index with variable radius(D2), dead individuals still have more stress than living individuals.

---

## [Decision Letter · Decision Letter 1]

11 Jun 2021

PONE-D-20-38825R1

Competition in Abies kawakamii forest at subtropical high mountain in Taiwan.

PLOS ONE

Dear Dr. Tzeng,

Thank you for submitting your manuscript to PLOS ONE. After careful consideration, we feel that it has merit but does not fully meet PLOS ONE’s publication criteria as it currently stands. Therefore, we invite you to submit a revised version of the manuscript that addresses the points raised during the review process.

A stated by reviewer #1, the main problem of your manuscript is terminology and language. I strongly recommend a linguistic revision.

We look forward to receiving your revised manuscript.

Kind regards,

Dusan Gomory

Academic Editor

PLOS ONE

Journal Requirements:

Additional Editor Comments (if provided):

Reviewers' comments:

Reviewer's Responses to Questions

**Comments to the Author**

1. If the authors have adequately addressed your comments raised in a previous round of review and you feel that this manuscript is now acceptable for publication, you may indicate that here to bypass the “Comments to the Author” section, enter your conflict of interest statement in the “Confidential to Editor” section, and submit your "Accept" recommendation.

Reviewer #1: All comments have been addressed

Reviewer #2: All comments have been addressed

2. Is the manuscript technically sound, and do the data support the conclusions?

Reviewer #1: Partly

Reviewer #2: Yes

3. Has the statistical analysis been performed appropriately and rigorously? 

Reviewer #1: Yes

Reviewer #2: Yes

4. Have the authors made all data underlying the findings in their manuscript fully available?

Reviewer #1: Yes

Reviewer #2: Yes

5. Is the manuscript presented in an intelligible fashion and written in standard English?

Reviewer #1: No

Reviewer #2: Yes

6. Review Comments to the Author

Reviewer #1: General comments

Authors revised the manuscript following most of the comments and suggestions provided in my previous review. Then this has improved the methods a little bit. However, authors did not improve the writing style and wording, and due to this, I am not convinced that the manuscript of the current version is still not suitable for publication. Authors need to do a lot, especially on choosing appropriate technical words, writing styles according to scientific writing, avoiding jargons and jumbo-mumbo words, and uncommon words, etc. Some forestry professional writer should rewrite this manuscript in order to make this publishable to the high impact journal like PlosOne. There are uncountable writing problems, among them I have pointed out some minor problems here.

Line 22: competition index is more appropriate term, rather than competitive index, so please change this accordingly here and other places as well. What do you mean by count plants? You might have measured tree attributes in sample plot and inter-tree distances. I do not see any necessity of counting trees and usefulness of this count number in your analysis.

Line 24-25: what stress resources? Give some example.

Line 28: Please specify those SSAs. Here you said SSAs, but in results and discussion, you have used the terms ecological attributes, please choose one and use consistently throughout the manuscript.

Line 31-32: What do you mean by analyzing the DBH? Maybe you are saying about Hegyi’s index based on DBH and distance. In addition to this, you also analyzed crown volumes and intertree distance, so please include this analysis here also.

Line 33-34: The findings indicate…… what do you mean by different results between CI and SSAs? Please specify. Don’t use term “would be” this indicates uncertainty, but your results are certain, so please use “was” instead of using “would be”

Line 48-50: What do mean by ethnic group? Use better terms. Please complete this sentence.

Line 63-64: I do not think you have determined the competitive distribution of species of interest. This needs mapping of vegetation based on competitive stresses they face.

Line 76: fir population….

Line 76-77: Not clear. You are not developing growth models, and your data does not allow doing so. Please this sentence more meaningful.

Line 79-80: Not sampling plot, but sample plot, please use this term consistently throughout the manuscript. Please also state hypotheses/questions that you would evaluate or answer in your study.

Line 87-88: Research site or study site? Please use the same terms consistently throughout the manuscript.

Line 116: plot size, but not plot’s size

Line 132-133: …for each target tree by using the Hegyi’s index based on DBH and crown volumes as shown in Eq. 1 and 2.

Line 137-140: Please each symbol, such as i and j, example Dj is DBH of competitor tree j, etc. Use the correct term consistently, such as competitor tree, but not competitive tree, throughout the manuscript.

Line 146-147: small case in We referenced ….. variable search

Line 155: what do you mean by woods? Please use “trees” instead of woods and other words consistently

Line 157-159: Please make this sentence more meaningful than what you have now. What do you mean by main forest trees on neighboring trees? You are using unnecessarily inappropriate words frequently. Please use standard and commonly used terms here and other places as well.

Line 160-167: this portion seems to be very poor, please use make correct wording and writing style. Because edge bias would underestimate the results of competition (32)……..we assume….. these plots are homogenous population…….. check carefully the meaning of these two statements. This paragraph should be more clearly rewritten in line with your methods.

Line 168: what do mean by different source? What are those sources? Not good wording at all.

Line 169-171: Again counting…..because of wording like this, your manuscript is not scientific article. Please choose appropriate word and write in scientific ways. What is nonparametric method? I do not see you have used any nonparametric methods.

Line 177-180: Also not good writing style here too. Please rewrite this part. How did you fit regression function, when did you fit, where is this function?

Line 184: Here ecological attributes, but in other places up to here, SSAs, why?

Line 185-212: Please be consistent. Please write results in past tense, and make discussion in present tense.

Reviewer #2: Dear Authors,

you have properly responded to all of my previous requests and concerns. The only thing that might be corrected before publication is the small part of the text that you have added in the process of revision. I have no further suggestions or requests.

7. PLOS authors have the option to publish the peer review history of their article (what does this mean?). If published, this will include your full peer review and any attached files.

Reviewer #1: No

Reviewer #2: No

---

## [Author Response · Author response to Decision Letter 1]

29 Jun 2021

The authors appreciate reviewers’ efforts and constructive comments. We have carefully considered all of the comments and thoroughly revised the manuscript. All revisions were highlighted in the revised manuscript, and a list of point-to-point responses was prepared in the following.

Comment from reviewer 1:

Comment 1. What’s stress resources, SSAs, woods, ethnic group?

Response: Thanks for the reviewer's comment. Those mentioned errors, we seriously have inspected the article structure and modified the most unclear sentence and technical words. We also improve the consistency of technical words and linguistic revised in the manuscript. For example, competitive index was revised to competition index. Redundant phrases are already deleted and reduce the description. We marked the track changes with red font in the revised manuscript.

Comment 2. Please also state hypotheses/questions that you would evaluate or answer in your study.

Response: Thanks for the reviewer’s comment. We state the five hypotheses about this study (Line 78-83) and the conclusion (Line 356-364) in the manuscript.

Comment 3. However, authors did not improve the writing style and wording, and due to this, I am not convinced that the manuscript of the current version is still not suitable for publication.

Response: Thanks for the reviewer’s comment. We deeply and sincerely modified the writing style of the manuscript and improved our description of data analyses (Line 135-176). We also add more relevant literature studies and citations (References 42-43, 47-48) to prove us assume. 

Comment from reviewer 2:

Comment 1. You have properly responded to all of my previous requests and concerns. The only thing that might be corrected before publication is the small part of the text that you have added in the process of revision. I have no further suggestions or requests.

Response: Thanks for the reviewer’s affirmation.

---

## [Editor Report · Decision Letter 2]

5 Jul 2021

Competition in Abies kawakamii forests at subtropical high mountain in Taiwan.

PONE-D-20-38825R2

Dear Dr. Tzeng,

We’re pleased to inform you that your manuscript has been judged scientifically suitable for publication and will be formally accepted for publication once it meets all outstanding technical requirements.

Kind regards,

Dusan Gomory

Academic Editor

PLOS ONE
---

## [Editor Report · Acceptance letter]

13 Jul 2021

PONE-D-20-38825R2 

Competition in *Abies kawakamii* forests at subtropical high mountain in Taiwan. 

Dear Dr. Tzeng:

I'm pleased to inform you that your manuscript has been deemed suitable for publication in PLOS ONE. Congratulations! Your manuscript is now with our production department. 

Kind regards, 

on behalf of

Dr Dusan Gomory 

Academic Editor

PLOS ONE